# Antibacterial Activity of Biosynthesized Copper Oxide Nanoparticles (CuONPs) Using *Ganoderma sessile*

**DOI:** 10.3390/antibiotics12081251

**Published:** 2023-07-29

**Authors:** Karla M. Flores-Rábago, Daniel Rivera-Mendoza, Alfredo R. Vilchis-Nestor, Karla Juarez-Moreno, Ernestina Castro-Longoria

**Affiliations:** 1Department of Microbiology, Center for Scientific Research and Higher Education of Ensenada (CICESE), Ensenada 22860, Mexico; dra.karlaflores@gmail.com (K.M.F.-R.); riveramd@cicese.edu.mx (D.R.-M.); 2Sustainable Chemistry Research Joint Center UAEM—UNAM (CCIQS), Toluca 50200, Mexico; arvilchisn@uaemex.mx; 3Center for Applied Physics and Advanced Technology, UNAM, Juriquilla 76230, Mexico; kjuarez@fata.unam.mx

**Keywords:** green method, copper oxide nanoparticles, antibacterial, ultrastructure

## Abstract

Copper oxide nanoparticles (CuONPs) were synthesized using an eco-friendly method and their antimicrobial and biocompatibility properties were determined. The supernatant and extract of the fungus *Ganoderma sessile* yielded small, quasi-spherical NPs with an average size of 4.5 ± 1.9 nm and 5.2 ± 2.1 nm, respectively. Nanoparticles were characterized by UV−Vis spectroscopy, transmission electron microscopy (TEM), Fourier transform infrared spectroscopy (FTIR), X-ray diffraction (XRD), dynamic light scattering (DLS), and zeta potential analysis. CuONPs showed antimicrobial activity against *Staphylococcus aureus* (*S. aureus*), *Escherichia coli* (*E. coli*), and *Pseudomonas aeruginosa* (*P. aeruginosa*). The half-maximal inhibitory concentration (IC50) for *E. coli* was 8.5 µg/mL, for *P. aeruginosa* was 4.1 µg/mL, and for *S. aureus* was 10.2 µg/mL. The ultrastructural analysis of bacteria exposed to CuONPs revealed the presence of small CuONPs all through the bacterial cells. Finally, the toxicity of CuONPs was analyzed in three mammalian cell lines: hepatocytes (AML-12), macrophages (RAW 264.7), and kidney (MDCK). Low concentrations (<15 µg/mL) of CuONPs-E were non-toxic to kidney cells and macrophages, and the hepatocytes were the most susceptible to CuONPs-S. The results obtained suggest that the CuONPs synthesized using the extract of the fungus *G. sessile* could be further evaluated for the treatment of superficial infectious diseases.

## 1. Introduction

Bacterial resistance and multi-resistance to antibiotics is a global public health problem that is continuously growing. In the last decades, nanotechnology has contributed to possible solutions for the management of bacterial resistance to antibiotics, and with new methods for the diagnosis and treatment of diseases [1,2,3,4,5]. The antimicrobial and antioxidant properties of metal and metal oxide nanoparticles (NPs) are promising for multiple applications in different areas of medicine [6,7,8,9]. It has been documented that their antibacterial effect to combat bacterial resistance to antibiotics is promising [10,11]. However, the cytotoxic effect on animal cells has limited the applications to combat resistant bacteria [12,13]. For this reason, in recent decades, alternative nanocomposites with outstanding biocompatibility, lower cytotoxicity, and immunogenicity have been sought [14,15].

The implementation of the biological methods, also called “green methods”, for the biosynthesis of metal and metal oxide NPs, have been extensively explored as they cause less damage to the environment compared with chemical and physical methods, and could improve the effectivity and biocompatibility of nanomaterials [16,17,18,19,20,21].

The use of fungi to synthesize NPs holds particular interest, as fungi secrete large amounts of enzymes and metabolites and are easier to manipulate in the laboratory [22]. For instance, the synthesis of metal and metal oxide NPs using fungi has been studied and reported for over two decades [22,23,24]. The intracellular compounds of the fungus as well as the compounds excreted into the medium (extracellular compounds) can be successfully used to produce NPs [25,26]. One of the benefits of using fungi for synthesizing metal-based NPs is that they do not require reducing chemical agents, protective agents, and/or stabilizing agents, which can be toxic [22]. In fact, the presence of bioactive metabolites acting as reducing and stabilizing agents may also reinforce the antioxidant and antimicrobial properties of NPs and reduce cytotoxic effects in animal cells [18,27,28].

For biomedical applications, silver and gold NPs have been the most studied; however, NPs such as copper oxide, iron oxide, and zinc oxide are currently being considered [29,30,31,32]. Metal oxide nanoparticles (MONPs) have attracted the attention of researchers due to their wide range of applications, and in the biomedical field they have high potential as biomedical materials. Besides their antimicrobial properties, MONPs could be employed in diagnosis, tissue therapy, wound healing, dentistry, immunotherapy, etc., and to treat lethal diseases by using them as drug delivery systems with minimal side effects [33]. Furthermore, green synthesized MONPs have excellent antioxidant and antimicrobial activities, and are less toxic compared with those that are chemically synthesized [18,28,34]. Recently, the production of copper and copper oxide NPs has gained interest, as in recent studies they have been shown to be useful for biomedical applications because of their anticancer, antioxidant, antimicrobial, and antidiabetic properties [9,18,33,35,36].

In this work, copper oxide nanoparticles (CuONPs) were produced using the extract and supernatant of *Ganoderma sessile*; this fungus is considered non-pathogenic to plants, animals, and humans; is easy to handle; and has a low production cost. Furthermore, mushrooms of the *Ganoderma* genus are considered medicinal, with antioxidant, anti-inflammatory, and antitumor properties [37,38]. In addition, it has been shown that the extract and the supernatant of liquid cultures of *G. sessile* are not toxic to mammalian cell lines [26]. The basidiocarp and mycelia of *Ganoderma* contain over 400 different bioactive compounds, which include mainly polysaccharides, triterpenoids, sterols, nucleotides, fatty acids, steroids, and proteins [39]. However, most of the research has been done with basidiocarps of *G. lucidum*. The bioactive compounds of *G. sessile* are still not reported, we are currently exploring which bioactive compounds may be present in the water-soluble fractions that we used for nanoparticle synthesis. Therefore, the main objective was to determine and compare the antibacterial properties and the biocompatibility of CuONPs synthesized using both the extract and the supernatant of liquid cultures, seeking alternative methods to combat human pathogens. To the best of our knowledge, this is the first report evaluating the potential use of *G. sessile* to produce MONPs. Here, we report an environmentally friendly and cost-effective synthesis of highly stable CuONPs with potential applicability in the biomedical field.

## 2. Materials and Methods

### 2.1. Strain, Media, and Growth Conditions

*Ganoderma sessile* was obtained from the fungal stock of the Microbiology Department (CICESE). The strain was inoculated in Petri plates with potato dextrose agar medium (PDA) and incubated at 30 °C for 96 h. Once the mycelium filled 75% of the culture dish, 10 plugs of the mycelium were taken with a standard size of 10 mm in diameter using a sterile borosilicate tube. The plugs were transferred to 250 mL Erlenmeyer flask with 100 mL of potato dextrose broth medium (PDB) and placed in an incubator with shaking at 120 rpm for 7 days at 30 °C [26].

### 2.2. Obtention of Fungal Supernatant and Extract

The obtention of the fungal supernatant and extract was done according to [26], with some modifications. To obtain the fungal supernatant, the biomass obtained from the liquid cultures was washed with sterile distilled water and placed in Erlenmeyer flasks with sterile deionized water in a 1:2 proportion (*w*/*v*) and incubated at 120 rpm for 24 h at 30 °C. After 24 h, the supernatant was obtained by filtration through a nitrocellulose membrane with a pore size of 0.45 µm (MF-Millipore) and subsequently through a nitrocellulose membrane with a pore size of 0.22 µm (MF-Millipore) to eliminate all biomass.

The fungal extract was obtained using biomass from liquid cultures in a 1:1 proportion (*w*/*v*); biomass was washed with sterile distilled water and macerated with sterile deionized water using an agate mortar. Once macerated, it was centrifuged for 15 min at 10,000 rpm at 22 °C, and finally, the aqueous extract was decanted. The extract obtained was filtered as described above.

### 2.3. Biosynthesis of Copper Oxide Nanoparticles (CuONPs)

CuONPs were synthesized according to [35], with some modifications. Synthesis was carried out using a fungal extract and/or supernatant mixed with pentahydrate copper sulfate (CuSO_4_∙5H_2_O) 5 mM (Sigma-Aldrich, St. Louis, MI, USA) as follows: copper sulfate and fungal supernatant (S) or extract (E) were mixed in different proportions according to [35], however the formation of NPs was observed only at the 3:1 proportion. A change in color (from clear to light blue/blue grey) was observed after the pH was adjusted to 10 with NaOH (10 mM), then the resulting suspension was incubated for 24 h at 60 °C.

### 2.4. Characterization of CuONPs

CuONPs were evaluated by UV−Vis spectroscopy at 200 to 800 nm [36]. A Perkin Elmer precisely UV−Vis lambda/25 spectrophotometer was used (PerkinElmer Inc., Waltham, MA, USA). Further characterization was carried out to determine the hydrodynamic diameter (HD), zeta potential (ZP), and polydispersity index (PDI) using a Zetasizer Nano ZS instrument (Malvern Panalytical Inc., Westborough, MA, USA). Then, 10 µL of nanoparticles were placed in carbon coated copper grids and analyzed under transmission electron microscopy (TEM) (Hitachi H7500, Hitachi Ltd., Tokyo, Japan) at 100 kV for size and shape determination. Additionally, 10 µL of CuONPs were placed in lacey carbon supported nickel grids and examined using a high-resolution transmission electron microscope (HRTEM) (JEM-2100 from JEOL, JEOL Ltd., Tokyo, Japan) operated at 200 kV. To assess the mean size of NPs, the ImageJ program (free version for Windows 1.8.0_172) was used.

The structural analysis was performed by X-ray diffraction (XRD) using a Bruker D8 Advanced diffractometer equipped with a Linxeye Detector. For this, CuONPs suspensions were centrifuged, then the pellets were collected and lyophilized to obtain a fine powder. The powder was placed into the sample holder and finally the XRD pattern was collected with Cu Kα radiation and a 2θ scanning angle variation between 10° and 80°. The phase analysis was supported with PDF-2 software.

To identify the functional groups that could be participating in the formation and stabilization of copper oxide nanoparticles, the samples were centrifuged, then the precipitates were collected and subsequently lyophilized to obtain a fine powder. The powder was placed into the sample holder and Fourier Transform Infrared Spectroscopic analysis (FTIR) was carried out. The FTIR spectra were acquired using a Bruker Tensor 27 spectrometer with total attenuated reflectance (FTIR-ATR) in a range of 4000 cm^−1^ to 400 cm^−1^ in transmittance mode.

### 2.5. Evaluation of Antibacterial Activity

The antibacterial activity of the synthesized CuONPs was evaluated against *Escherichia coli* (ATCC 25922), *Staphylococcus aureus* (ATCC 25923), and *Pseudomonas aeruginosa* (ATCC 27853) as the minimum inhibitory concentration (MIC). For this, the plate microdilution assay was used, following the protocol of the Clinical and Laboratory Standards Institute [40]. For the microdilution assay, each bacterial strain was inoculated in Petri dishes with LB agar medium at 37 °C for 24 h. Subsequently, four colonies were inoculated in Mueller Hinton broth (MHB) and incubated at 37 °C under agitation, until reaching an approximate concentration of 5 × 10^5^ CFU/mL [41]. Then, the MIC determination was carried out in polystyrene 96-well microplates (Costar 3595). First, MHB was mixed with CuONPs to prepare the highest concentrations of CuONPs-S and CuONPs-E tested (26.5 and 27.5 μg/mL, respectively); then, serial dilutions were carried out to obtain the different concentrations used. Finally, in each well, 5 µL of the bacterial inoculum was added. MHB without inoculation and MHB with bacterial inoculum without CuONPs were used as the controls. The plates were incubated at 37 °C for 24 h. For the analysis, the Multiskan Sky version 1.00.55 plate reader (Thermo Fisher Scientific™, Waltham, MA, USA) was used, with Skanlt 6.0.1 software. After the incubation time, 10 µL of each well was inoculated in Petri dishes with LB agar and incubated for 24 to 48 h at 37 °C to determine the bacterial growth (% CFU); the assays were done in triplicate.

### 2.6. ROS Production in Bacteria

To evaluate the production of reactive oxygen species (ROS) in bacteria, the protocol described in Garcia-Marin et al., 2022 [42], with some modifications, was carried out, as follows: in a 96-well plate, cells were seeded at 1 × 10^5^ cells per well, and exposed to different CuONPs concentrations (19.9, 9.95, 4.97, 2.48, 1.24, and 0.62 μg/mL), followed by incubation at 37 °C for 24 h. For this preparation, we used a stock suspension of CuONPs at 39.8 μg/mL. First, 100 μL of the NPs stock were mixed with 100 μL of each bacterial suspension in MHB; in this way, the NP concentration was 19.9 μg/mL. For the following NP concentrations, we completed serial dilutions to obtain all concentrations previously mentioned. Bacteria incubated with 1 mM of H_2_O_2_ were considered as a positive control, and those without CuONPs were used as the negative control cells. After treatment with CuONPs, the cells were washed thrice with 200 μL of PBS 1×. Then, PBS was removed and the cells were incubated at 37 °C in darkness with 100 μL of DCFDA (20,70-dichlorofluorescein diacetate, 45 μM) (D6883 Sigma-Aldrich) for 60 min. Fluorescence (λex = 485 nm and λem = 520 nm) was measured using a Cary Eclipse fluorescence spectrophotometer (Agilent Technologies, Santa Clara, CA, USA).

### 2.7. Ultrastructural Analysis of Bacteria

The interaction of CuONPs with bacteria was analyzed for Gram-negative *E. coli* and Gram-positive *S. aureus*. Strains were grown in MHB with CuONPs at the MIC obtained for each strain, and control cultures were also prepared using the following protocol: After incubation for 24 h, the cells were fixed with 2.5% glutaraldehyde in 0.05 M sodium phosphate for 30 min at ambient temperature. Then, cells were post-fixed with 2% OsO_4_ at 4 °C for 2 h. Subsequently, samples were dehydrated in ethanol series and then infiltrated in Spurr resin/ethanol according to [43]. Samples were then polymerized at 60 °C for 24 h; afterwards, sections of 70 nm thick were obtained in a Leica Ultracut-R ultramicrotome (Leica Microsystems Inc., Buffalo Grove, IL, USA). The samples were mounted in formvar/carbon 75 mesh copper grids and analyzed under TEM (Hitachi H7500, Hitachi Ltd., Tokyo, Japan), operated at 100 keV. For better CuONPs detection, the sections were not post-stained.

### 2.8. Biocompatibility of CuONPs in Mammalian Cell Lines

The effect of CuONPs in cell lines was evaluated in the Madin-Darby canine kidney cell line (MDCK), a macrophage cell line (RAW 264.7), and a hepatocyte cell line (AML-12). To determine the susceptibility of the different cell lines to CuONPs, similar concentrations used for bacterial inhibition were used. A 96-well microplate was used, with each well containing 10,000 cells in a final volume of 100 µL. The cells were incubated at 37 °C with 5% CO_2_ for 24 h. After incubation, the culture medium was discarded and the cells were exposed to different volumes of CuONPs (22.2, 11.1, 5.6, 2.8, 1.4, and 0.7 µL), obtained with the aqueous extract and supernatant of *G. sessile*. The final volume of the wells was adjusted to 100 µL with supplemented DMEM medium, and the plate was incubated at 37 °C with 5% CO_2_. DMEM culture medium without CuONPs was used as a positive control, and 1% Triton X-100 in PBS was used as a negative control. After 24 h of incubation with CuONPs, the culture medium was discarded, and the cultures were washed three times with 200 µL of PBS. To determine cell viability, the colorimetric method of the reduction of (3-[4,5-dimethylthiazol-2-yl]-2,5 diphenyl tetrazolium bromide) (MTT) was used (Sigma Aldrich M-8910) [44]. The absorbance was read with a UV−Vis spectrophotometer (Thermo Scientific Multiskan GO) at 570 nm and 690 nm. All MTT reduction assays were performed independently in triplicate.

### 2.9. Statistical Analysis

Statistical analysis was performed using the GraphPad Prism 9.3.0 software. The average size of NPs was calculated measuring 1000 NPs of each sample. Plotted data were reported as mean ± standard deviation. Two-way ANOVA followed by a Tukey test was used to detect significant differences in the mammalian cell viability assays.

## 3. Results

### 3.1. Nanoparticle Characterization

After incubation, CuONPs were firstly evaluated by UV−Vis spectroscopy. Synthesized NPs using the supernatant (CuONPs-S) displayed a dark blue grey suspension, and the absorbance peak was observed at 290.73 nm (Figure 1A). Nanoparticles were polydisperse (PDI = 0.619), quasi-spherical (Figure 1B), with a size range of 1–15 nm, a mean size of 4.5 ± 1.9 nm (Figure 1C), and zeta potential of −28.7 mV (Figure 1D). The negative value of the Z-potential can be attributable to the nature of the biomolecules involved in the stabilization of the nanoparticles; a large negative value of Z-potential usually indicates high stability [45] of the suspension due to electrostatic repulsion between nanoparticles.

By using the extract (CuONPs-E), a light blue suspension was obtained, the absorbance peak was observed at 296.10 nm (Figure 2A), and the blue color was associated with Cu^+2^ species. Nanoparticles were polydisperse (PDI = 0.674), quasi-spherical, and seemed to be embedded in a matrix of organic matter (Figure 2B), showing a size range of 1 to 20 nm and a mean size of 5.2 ± 2.1 nm (Figure 2C). The CuONPs-E nanoparticles exhibited −24.8 mV of zeta potential (values of ± 30 mV are considered highly stable [45]), which can be associated with the capping biomolecules that surround the nanoparticles by polar functional groups with negative charge (Figure 2D).

HRTEM images in Figure 3 correspond to CuONPs-S (Figure 3A) and CuONPs (Figure 3B), respectively. It is noticeable that copper oxide nanoparticles were embedded in the biomass, suggesting that biomolecules, such as residual proteins and amino acids from the fungus, act as a capping agent of the CuONPs. The lattice spacing in both micrographs (Figure 3A,B) was about 0.23 nm between the (111) plane, consistent with the monoclinic structure of the CuO (JCPDS Card no. 00-048-1548). The insets are the SAED patterns, which reveal the polycrystalline nature of the sample with (111) and (20$\bar{2}$) planes and diffuse rings that could be attributable to the presence of capping biomolecules.

### 3.2. X-Ray Diffraction (XRD) Patterns of Synthesized CuONPs

The X-ray diffraction patterns of the biosynthesized CuONPs using the supernatant and extract of *G. sessile* are shown Figure 4. The noisy profiles of both patterns are usually associated with amorphous materials, in this case, the biomolecules, such as residual proteins of *G. sessile* capping the CuO nanoparticles. The Bragg reflections show that the two samples have the same space group C2/c related with the monoclinic system of CuO. The peaks at 2θ = 35.5°, 38.7°, and 48.7° were assigned to diffraction from the (11$\bar{1}$), (111), and (20$\bar{2}$) planes of the monoclinic structure of copper(II) oxide, which is in good agreement with JCPDS Card No. 00-048-1548. Furthermore, additional reflections were detected, although they exhibited a low intensity signal in contrast with the CuO phase; this can be associated with Cu(NO_3_)_2_ and the copper complex [Cu(OH)_2_NH_3_H_2_O] on the base of JCPDF cards no. 00-019-0414 and 00-042-0637, respectively.

### 3.3. FTIR Analysis of Synthesized CuONPs

Figure 5 shows the FTIR spectra of the CuONPs, which exhibit bands around 3387 and 3268 cm^−1^ associated with the characteristic appearance of compounds with N–H and OH groups, respectively [46]. Weak bands at 2972 and 2879 cm^−1^ are characteristic of stretching vibrations of the methyl groups [47]. The peak at 1077 cm^−1^ spectrum was attributed to primary amines [47], which confirmed the presence of residual amino acids and proteins from the fungus, which are involved in the synthesis and stabilization of the copper oxide nanoparticles. The fungal supernatant and fungus extract were expected to contain amino acids, enzymes, and residual proteins, which might act as stabilization agents of CuONPs. Both spectra (Figure 5) showed a band at 471 cm^−1^ that corresponded to the stretching vibration of the bond Cu–N [48], indicating a strong interaction between the nanoparticles and the fungal biomolecules.

### 3.4. Antibacterial Capacity

Preliminary results using the disk diffusion assays showed inhibition of bacteria exposed to CuONPs for both Gram-negative and Gram-positive bacteria (Figure 6A shows the representative image). No inhibition was observed using CuSO_4_ and, as can be observed, some bacteria grew within the inhibition zone, possibly due to a diffused lower concentration. Nevertheless, low concentrations of CuONPs were required for bacterial inhibition, where IC50 was 10.2 and 8.8 µg/mL for *S. aureus*, 8.5 and 8.0 µg/mL for *E. coli*, and 4.1 and 3.4 µg/mL for *P. aeruginosa* (Table 1). As can be observed, similar results were obtained when using both types of NPs, but *P. aeruginosa* was the most susceptible to both treatments (Figure 6). Although the MIC for bacterial strains was similar, slightly higher concentrations of CuONPs-E were required. The MIC obtained for *E. coli* was 15.9 µg/mL with CuONPs-S and 16.5 µg/mL using CuONPs-E. For *P. aeruginosa,* an MIC of 13.7 µg/mL of CuONPs-S and 16.5 µg/mL of CuONPs-E was obtained. However, at the highest concentrations tested, some colonies of *S. aureus* were present (Figure 6B).

### 3.5. ROS Production in Bacteria

ROS production was measured to determine if exposure to CuONPs activated a stress response of oxidation in bacteria. We observed that only at the highest concentration of NPs tested (close to the MIC obtained) was the overproduction of ROS higher (Figure 7). As shown in Figure 7, the ROS production at 0.62 μg/mL and 1.24 μg/mL for CuONPs and the negative control was similar. Comparable stress was detected at 9.95, 4.97, and 2.48 μg/mL. In the case of *E. coli*, we can observe that at the highest concentration of NPs (19.9 μg/mL), the response was similar to the positive control. In general, *S. aureus* had a lower production of ROS at all CuONPs concentrations.

### 3.6. Ultrastructural Analysis of Bacteria

To assess the interaction of CuONPs with bacterial cells, *E. coli* and *S. aureus* were exposed to MIC and analyzed under TEM. At low magnifications, it was difficult to observe NPs because their mean size was about 5 nm. However, at higher magnifications, it was possible to detect that in all cases, CuONPs internalized into the cells. No specific accumulation was observed for *E. coli* (Figure 8A,B) or *S. aureus* (Figure 8C,D). However, NPs were detected throughout the cell and occasionally small accumulations were observed (Figure 8A,B). Control cultures were also analyzed to discard that any other material could be accumulated in the cells treated with NPs and they were observed without any accumulation of foreign material (Figure 8E–H). The low contrast in cells was because no post-staining was used to easily detect small NPs.

### 3.7. Toxicity of CuONPs in Mammalian Cell Lines

To determine the cytotoxicity of the biosynthesized CuONPs, canine kidney (MDCK), murine macrophages (RAW 264.7), and murine hepatocytes (AML-12) were exposed to different volumes of CuONPs, and the cell viability was determined and reported at calculated final concentrations of CuONPs. As shown in Figure 9, cell viability was dose-dependent, and the CuONPs-S were more cytotoxic than CUONPs-E. In addition, the cell viability of hepatocytes treated with CuONPs-S were the most susceptible, showing acceptable cell viability only at concentrations lower that 3.6 µg/mL; while using higher concentrations of CuONPs-E, cell viability was higher at most concentrations used (Figure 9B). In the case of macrophages treated with CuONPs-S, a decrease in cell viability was also observed, proportional to the increase in the concentration of CuONPs-S, similar to that observed in hepatocytes. Macrophages treated with concentrations of 0.9 to 15.26 µg/mL of CuONPs-E did not induce any change in cell viability; however, at a concentration of 30.5 µg/mL of CuONPs-E, we observed a significant decrease of 14% in cell viability (Figure 9B). For kidney cells, good cell viability was observed at concentrations up to 7.3 µg/mL CuONPs-S (Figure 9A), while at concentrations of 14.7 and 29.5 µg/mL, the cell viability of kidney cells decreased drastically. However, excellent cell viability was observed in cells exposed to CuONPs-E, at concentrations between 0.9 µg/mL to 15.2 µg/mL of CuONPs; only at the highest concentration of 30.5 µg/mL was the percentage of viability decreased to 55% (Figure 9).

To assess whether the cytotoxic effect observed in cell lines exposed to CuONPs-S and CuONPs-E was merely due to the NPs and not to the aqueous extract or supernatant of *G. sessile*, it was decided to compare the viability of hepatocytes, macrophages, and kidney cells in the presence of two volumes of S and E—the lowest (0.7 µL) and the highest (22.2 µL) used for CuONPs, as well as the NPs synthesized from them. In the case of the cell lines treated with the supernatant, a slight decrease in cell viability was observed compared with the control, as shown in Figure 10. On the contrary, in hepatocytes and macrophages treated with the extract, an increase in cell viability was observed compared with the control, whereas in kidney cells, cell viability remained like the control.

## 4. Discussion

In this work, we report, for the first time, the synthesis of MONPs using *G. sessile*, which is considered as a non-photogenic fungus; in fact, members of the *Ganoderma* family are considered as medicinal [49]. It is important to mention that the pH adjustment is crucial for the biosynthesis of CuNPs and CuONPs [35,50,51,52]. For example, Noor et al. [35] reported the pH adjustment for the synthesis of CuNPs using the extract of the fungus *Aspergillus niger* with CuSO_4_. It was noted that when adjusting the pH to 5, 7, and 8, the synthesis of CuNPs was only achieved under a pH of 7. In the green synthesis of CuONPs using *Galphimia glauca* leaf and flower extracts, the authors observed that the most favorable pH for the synthesis of CuONPs was 12. At a low pH, they reported that the activity of the carboxyl groups present in the extract of *G. glauca* decreased in such a way that with a pH of 2, they obtained larger spherical CuONPs (50–60 nm) with a greater tendency to agglomeration [52]. The need to adjust the pH may be due to the decreased activity of reducing biomolecules due to the acidity of the precursor salt and/or the extract during the synthesis process. In this work, the pH of the 5 mM CuSO_4_·5H_2_O solution was 3.72, while the pH of the mixture of the supernatant and extract of *G. sessile* with CuSO_4_·5H_2_O, was 3.18 and 3.37, respectively (ratio 1:3). Therefore, for the synthesis of CuONPs using the aqueous extract and the supernatant of *G. sessile*, it was necessary to adjust the pH to 10.

The obtained CuONPs were analyzed using spectrophotometry in the UV−Vis light range, and an absorbance peak was found at 290.73 nm using the supernatant and 296.10 nm using the extract of *G. sessile*, similar to the absorbance peak at 290 nm reported using the *Camellia japonica* leaf extract [53].

Regarding the sizes of the CuONPs, it was found that those obtained by TEM differed considerably from the hydrodynamic diameters obtained with DLS, which may be due to the adhesion of components (molecules and proteins) present in the extracts used from *G. sessile*. In addition, this material serves as a capping agent, which maintains the stability of NPs, evidenced by the zeta potential values (−28.7 and −24.8 mV for CuONPs-S and CuONPs-E, respectively).

The obtained nanoparticles were quasi spherical in shape; HRTEM and SAED analysis revealed their polycrystalline nature. The diffractograms revealed the presence of three copper compounds and both types of nanoparticles seemed to have a considerable amorphous component, especially NPs made with the extract (E); this is expected from the biomass residues from the intracellular components. The copper(II) oxide signals, in NPs from the supernatant (S), were clearly visible; in NPs from the extract (E), they were just resolved. The additional signals belonged to copper nitrate and there were a couple of signals at low angles that seemed to coincide with a complex of Cu with OH and NH_3_, which could be due to the interaction of Cu^2+^ with OH and amino groups of the extract.

The profile of the FTIR spectra in the region between 3500 and 3250 cm^−1^ had the characteristic appearance of compounds with N–H, either ammonia or amines; this bond can be formed between Cu and protein residues. In fact, these results coincide with what was found in the DRX analysis.

The antibacterial activity of the CuONPs was determined through the bacterial growth inhibition assay; the MIC obtained was 16.5 µg/mL for *E. coli* and *P*. *aeruginosa* when using CuONPs-E. It is important to mention that both types of CuONPs were stable and maintained their antibacterial capacity for 2.5 years at ambient conditions. The concentrations needed for bacterial inhibition can be considered low, as higher concentrations of copper NPs for *E. coli* inhibition were reported by [54]; the MIC obtained for different strains were in a range of 140 to 280 µg/mL for *E. coli*, 140 µg/mL for *S. aureus* and 20 µg/mL for *Bacillus subtilis*. In that study, the authors found that the Gram-positive *B. subtilis* strain MTCC 441 was more sensitive to the copper NPs than the silver NPs. Biologically synthesized CuONPs could show different effectiveness against some bacterial strains, possibly depending on the bioactive metabolites acting in a synergic way. For instance, Nagore et al., [55] reported an MIC of 25 µg/mL for *E. coli,* while in another study, it was reported that CuONPs were not effective for *E. coli* at concentrations higher than 200 µg/mL [36]. In our study, although CuONPs significantly reduced *S. aureus* growth, they were more effective against *P. aeruginosa* and *E. coli* at concentrations of 16.5 μg/mL.

Some of the proposed mechanisms of the antibacterial effect are through the interaction and disruption of the bacterial cell membrane, which allows for the loss of cytoplasmic content [56,57]. In our study, although no evident specific accumulation of CuONPs was detected, in *E. coli* and *S. aureus,* a slight accumulation in the cell wall and cytoplasmic membrane was found, this contrast with the results of [29]. They reported that the ultrastructure of *B. subtilis* and *P. aeruginosa* remarkably changed after exposure to cuprous oxide NPs. They found a high accumulation of NPs attached to the surface of *B. subtilis*, displaying low density regions due to the permeability of the cell wall and leakage of the cytoplasmic content. However, bacteria were exposed to higher concentrations and different types of NPs; lipopeptide-stabilized Cu_2_ONPs with a higher average size (30 ± 2 nm) were used [29].

The antibacterial activity of CuNPs evaluated in *E. coli* and *Proteus vulgaris* revealed a dose-dependent bactericidal action. Exposure to NPs provoked ROS generation, loss of membrane permeability, and leakage of cytoplasmic components, finally causing bacterial cell death [57]. The antibacterial mechanisms of Cu/CuONPs include the damage of cell membrane, generation of ROS, destabilization of ribosomes, dysfunction of mitochondria, lipid peroxidation, protein oxidation, and DNA degradation [58,59].

Regarding the oxidative stress that nanometals could cause to bacteria, it is well documented that elevated ROS production occurs when bacteria is exposed to high concentrations of AgNPs and Cu/CuONPs [57,58,60,61,62]. The same was true in our study; although we used relatively low concentrations of CuONPs, we detected elevated ROS production in all bacteria only at the highest concentration of CuONPs tested.

It is possible that ROS overproduction at relatively low concentrations of CuONPs is associated with the material used to biosynthesize our NPs; this could be related to the ability of organic molecules (present in the supernatant/extract) to interact with cell membranes. We need to investigate further the hydrophilicity/hydrophobicity interactions between molecules contained in the fungal extracts used herein, which could be crucial for overpassing the natural barriers in cells. Indeed, ions produced by metals can interact with some electrons in the molecules of cells membranes, but it is probably that a synergic effect is produced by using the organic molecules in the extract and supernatant of *G. sessile* that provide the NPs a surface corona that can interact with biological systems [63,64,65].

Nanometal oxides such as ZnO and CuO NPs synthesized by a sol–gel combustion route were reported as excellent antibacterial agents against both Gram-positive and Gram-negative bacteria. However, the authors stated the importance of identifying the key physicochemical properties of nanometal oxides that govern antibacterial capacity and cytotoxicity to mammalian cells [66].

In this respect, cellular toxicity produced by metal oxide NPs is well documented [12,67,68]. However, the specific mechanisms of this toxicity are not yet fully described. There are different theories of the toxic effect on animal cells, one of them is based on the production of reactive oxygen species (ROS) as one of the determining factors of cell death [52]. In this way, in the biomedical area, metallic NPs can act as a therapeutic agent and consider this toxic property as an undesired effect. Nevertheless, this toxic effect could be useful to limit and control highly lethal cancers. However, in some of the biomedical applications of CuONPs and other metallic NPs, it is necessary to focus on minimizing toxicity. In such cases, CuONPs can be embedded in materials for medical use, providing the antibacterial effect only on contact with the surface, but without generating toxicity in those who handle it [69,70]. To reduce toxicity, in this study, we used fungal supernatant and extract of a non-pathogenic fungus, which proved to have no-toxic response in three mammalian cell lines (Figure 9). Nevertheless, CuONPs-E were less toxic for mammalian cells lines with IC50 of 29.5 µg/mL for macrophages and kidney cells, and for hepatocytes the IC50 was 14.7 µg/mL, close to the MIC found for bacteria (16.5 µg/mL). Thus, these concentrations of Cu could be considered safe for humans as the World Health Organization (WHO) reported a value of 2.0 mg/L as a regulation or guideline for copper in drinking water [71]. In addition, only at high concentrations of Cu have animal studies shown liver injury and inflammatory responses to Cu administered above 4 mg/kg/day [72].

It is important to identify the degree of toxicity that CuONPs can produce, regardless of the synthesis method to produce them. Determining the toxic concentrations of CuONPs will allow for knowing the safe concentrations for future applications in the biomedical area. In this work, the CuONPs synthesized with the extract proved to be less toxic than the CuONPs synthesized with the supernatant of *G. sessile*, allowing for the appropriate selection for future applications.

## 5. Conclusions

CuONPs were successfully obtained using the extracellular metabolites (supernatant) and the intracellular components (extract) of the fungus *G. sessile*. The antibacterial property against pathogenic bacteria makes them a possible eco-friendly alternative for managing infectious diseases as CuONPs were stable and maintained their antibacterial capacity for 2.5 years at ambient conditions. Cell viability results indicate that low concentrations (<15 µg/mL) of CuONPs are not toxic to kidney and macrophage cell lines and have an IC50 of 14.7 µg/mL for hepatocytes. This demonstrates the biocompatibility of the biosynthesized CuONPs and makes them excellent candidates for the treatment of superficial infectious diseases. However, it would be necessary to demonstrate that they have a low absorption into the systemic circulation, and consequently a low concentration during renal excretion. Ongoing work includes the identification of the water-soluble bioactive metabolites present in the extract and supernatant of *G. sessile*. The identification of bioactive metabolites attached to NPs would be necessary when using the green synthesizing methods, and to determine if they act in a synergistically way. One challenge using the green approach would be to design nanomaterials with a broad range of antibacterial or antifungal properties with low or no adverse effects in animal cells.

## Figures and Tables

**Figure 1 antibiotics-12-01251-f001:**
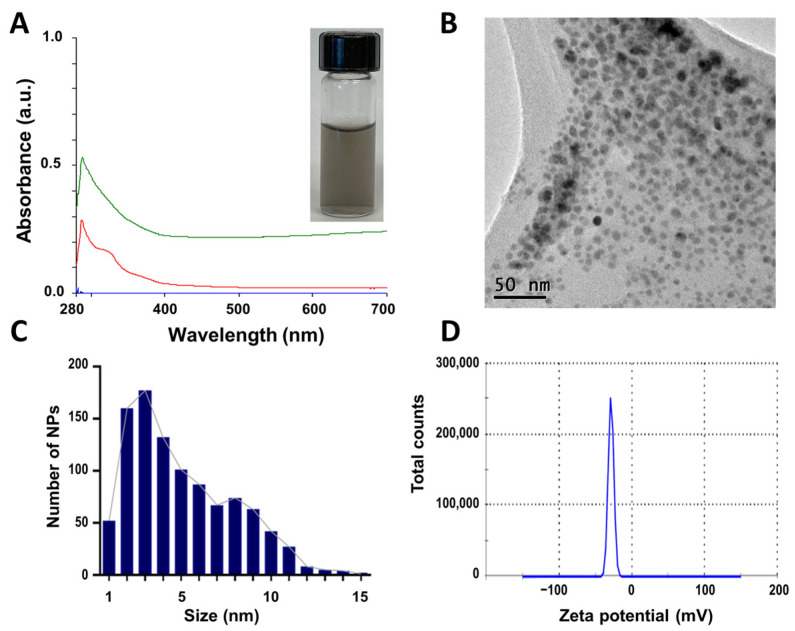
Characterization of CuONPs obtained with the supernatant of *G. sessile* (CuONPs-S). (**A**) UV−Vis spectroscopy curves of CuONPs-S (green line) and supernatant (red line), inset shows nanoparticle suspension. (**B**) TEM image of CuONPs-S showing a quasi-spherical shape. (**C**) Size distribution histogram of CuONPs-S. (**D**) Zeta potential.

**Figure 2 antibiotics-12-01251-f002:**
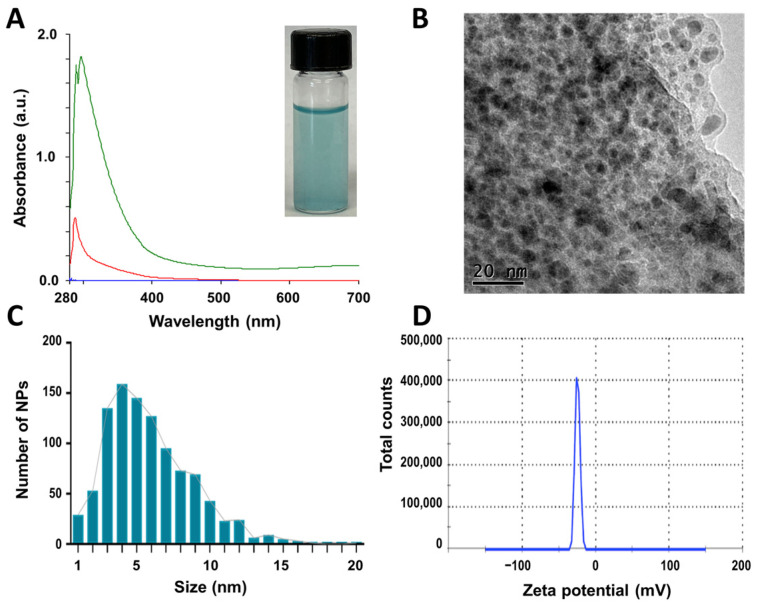
Characterization of CuONPs obtained with the extract of *G. sessile* (CuONPs-E). (**A**) UV−Vis spectroscopy curves of CuONPs-E (green line) and supernatant (red line), inset shows nanoparticle suspension. (**B**) TEM image of CuONPs-E showing quasi-spherical shape and matrix of organic matter. (**C**) Size distribution histogram of CuONPs-E. (**D**) Zeta potential of the CuONPs.

**Figure 3 antibiotics-12-01251-f003:**
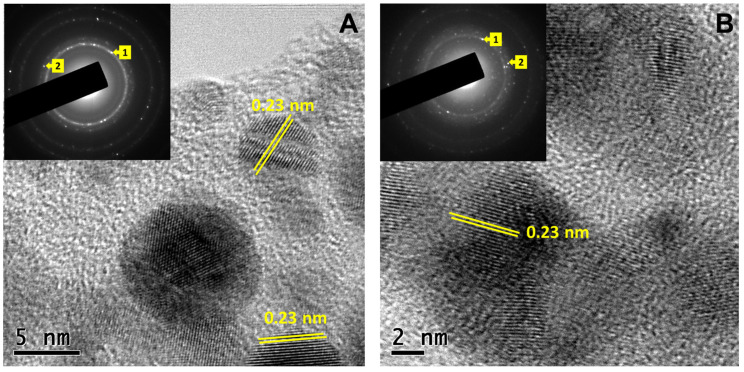
HRTEM images of CuONPs obtained with the supernatant and extract of *G. sessile*. (**A**) CuONPs-S, (**B**) CuONPs-E. The insets show their corresponding SAED patterns; 1 and 2 are associated with the (111) and (20$\bar{2}$) planes.

**Figure 4 antibiotics-12-01251-f004:**
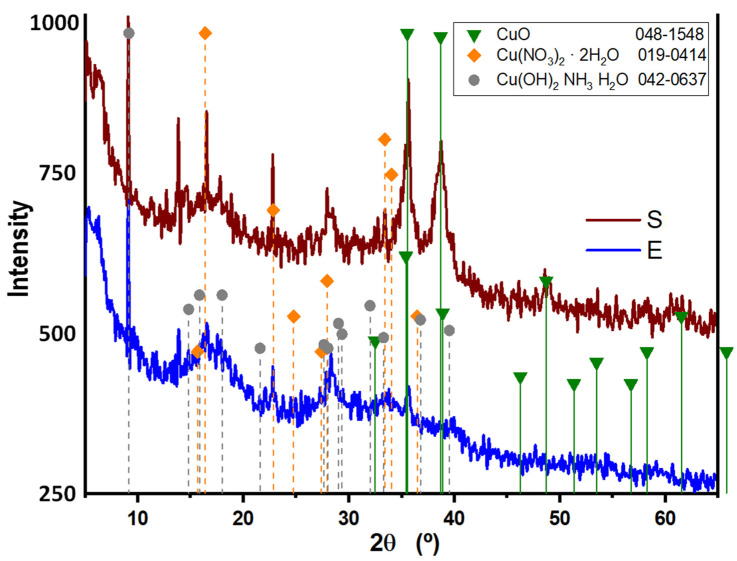
XRD spectrum of copper oxide nanoparticles obtained with the supernatant (S) and extract (E) of *G. sessile*.

**Figure 5 antibiotics-12-01251-f005:**
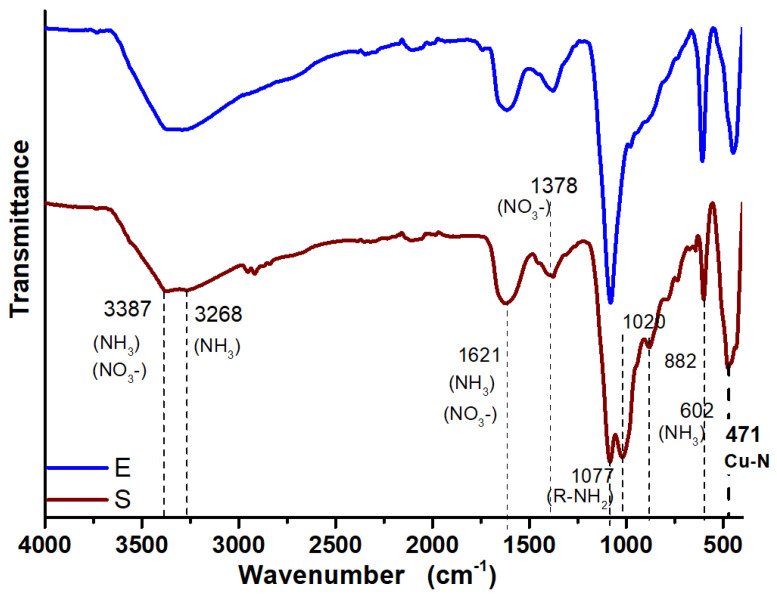
FTIR spectra of CuONPs synthesized with the supernatant (S) and extract (E) of *G. sessile*.

**Figure 6 antibiotics-12-01251-f006:**
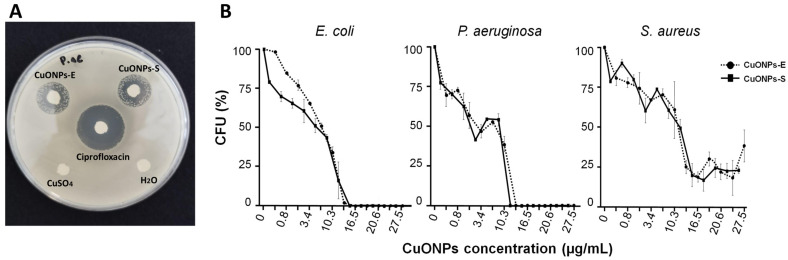
Antibacterial activity of CuONPs-S and CUONPs-E. (**A**) Disk diffusion method assay. (**B**) Percentage of colony formation of bacteria recovered after treatment with CuONPs for 24 h.

**Figure 7 antibiotics-12-01251-f007:**
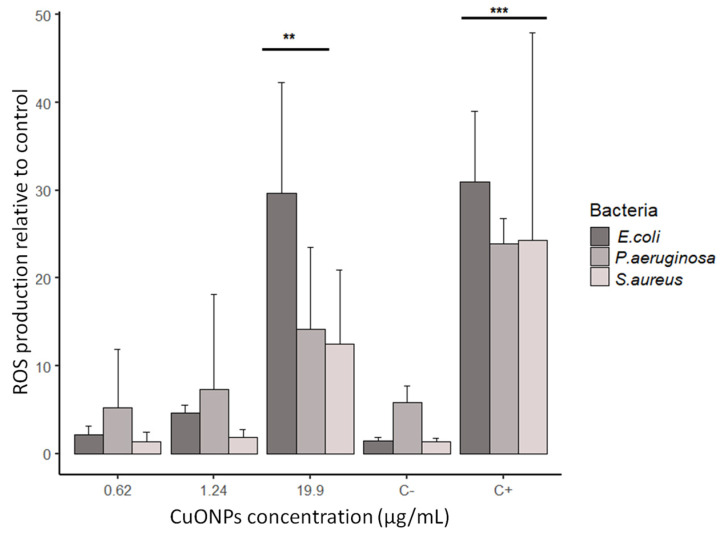
ROS production by bacteria exposed to CuONPs-S synthesized using *G. sessile*. C− = no treatment; C+ = positive control using 1 mM H_2_O_2_. Results are mean + SD (n = 3), ** *p* < 0.01; *** *p* < 0.001 (two-way ANOVA with a Dunnett’s test).

**Figure 8 antibiotics-12-01251-f008:**
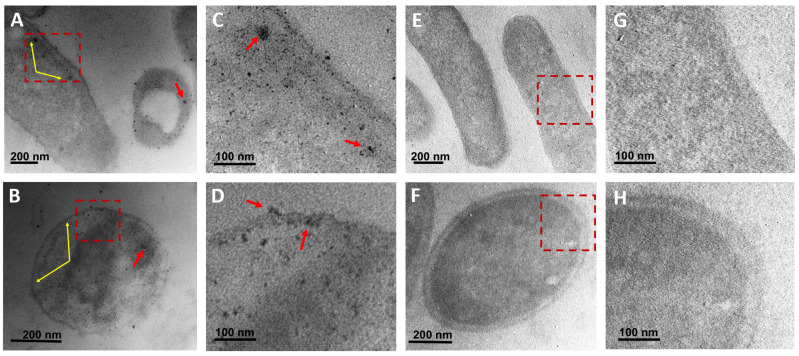
Ultrastructural analysis of bacteria exposed for 24 h to CuONPs biosynthesized using *G. sessile*. (**A**,**B**) *E. coli* and *S. aureus* showing a high internal number of small NPs, respectively. (**C**,**D**) Amplification of marked area in (**A**,**B**) showing in more detail the presence on NPs. (**E**,**F**) Cells of *E. coli* and *S. aureus* from the control cultures, without NPs treatment. (**G**,**H**) Amplification of marked areas in (**E**,**F**) showing no accumulation of any material. Red arrows indicate small accumulation of NPs, yellow arrows point out at slight accumulation of NPs within the cell wall and at the cell membrane.

**Figure 9 antibiotics-12-01251-f009:**
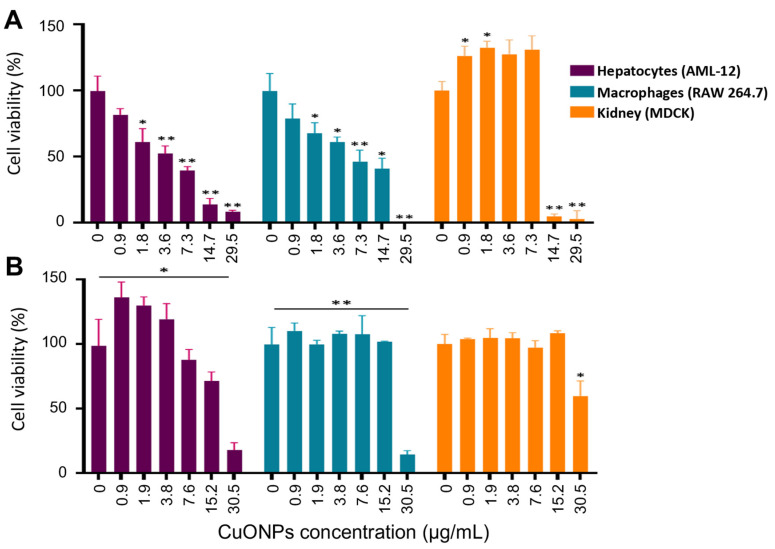
Cell viability assay in cell lines exposed to CuONPs-S (**A**) and CuONPs-E (**B**). Bars represent the mean + SD. * *p* ≥ 0.05, ** *p* ≥ 0.01.

**Figure 10 antibiotics-12-01251-f010:**
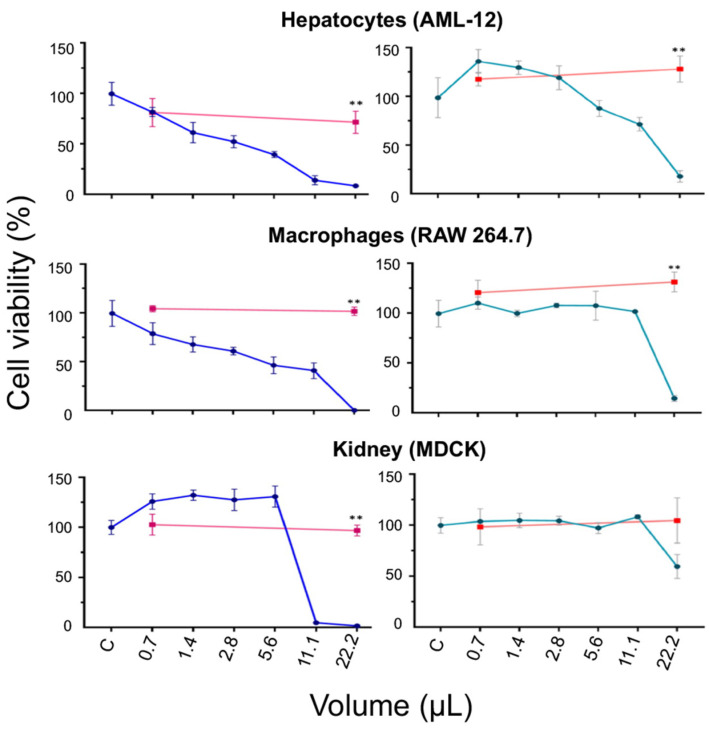
Cell viability assay in cell lines exposed to CuONPs-S (blue line) and CuONPs-E (cyan line). Red lines represent cell viability exposed to the fungal supernatant (**left side**) or extract (**right side**), respectively. ** *p* ≥ 0.01.

**Table 1 antibiotics-12-01251-t001:** Physicochemical characteristics, antimicrobial effect, and toxicity analysis of CuONPs synthesized with the supernatant (S) and extract (E) of *G. sessile*.

NPs	Physicochemical Characteristics				Antimicrobial Effect						Toxicity		
	TEM		PDI	ZP (mV)	Bacteria (µg/mL)						Cell Lines		
	Size (nm)	Shape			*E. coli*		*P. aeruginosa*		*S. aureus*		AML-12	RAW-264.7	MDCK
					MIC	IC50	MIC	IC50	MIC	IC50	IC50	IC50	IC50
CuONPs-S	4.5 ± 1.9	QS	0.619	−28.7	15.9	8.0	13.7	4.1	ND	8.8	3.6	7.3	7.3
CuONPs-E	5.2 ± 2.1	QS	0.674	−24.8	16.5	8.5	16.5	3.4	ND	10.2	14.7	29.5	29.5

NPs: nanoparticles; TEM: transmission electron microscopy; QS: quasi-spherical; PDI: polydispersity index; ZP: zeta potential; ND: not determined.

## Data Availability

Not applicable.

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
