# Peer review of "Antibacterial Activity of Biosynthesized Copper Oxide Nanoparticles (CuONPs) Using Ganoderma sessile"

_antibiotics, 2023, doi:10.3390/antibiotics12081251_

Round 1

Reviewer 1 Report

The manuscript entitled “Antibacterial activity of biosynthesized copper oxide nanoparticles (CuONPs) using Ganoderma sessile" has reported formation and characterization of CuONPs using Ganoderma sessile fungal supernatant and pellet. Manuscript is technically well-structured, and under the scope of the journal and will be of interest to the readers.

Few queries/suggestions are:

1.      What is % formation of nanoparticles (i.e. what amount of salts are converted to nanoparticles)?

2.      Have you processed the nanoparticles after preparation to remove the CuSO4 salts? If not, then have you tested the antimicrobial activity of CuSO4 salts as control?

3.      The Size distribution histogram is showing AgNPs instead of CuONPs  (fig 1&2 C)

4.      UV-Vis peak of CuONPs is also not clear as both supernatant (red) and CuONPs (green) showing same peak (Fig1&2 A) (no shift in the peak)

5.      E. coli growth is inhibited at 16.5ug/mL but again growth at 27.5 ug/mL. What could be reason for that?

6.      What is reason for the fluctuation in growth at different concentration of CuONPs in P. aeruginosa and S. aureus?

Author Response

Reviewer 1

1. What is % formation of nanoparticles (i.e. what amount of salts are converted to nanoparticles)?

The exact answer to this question would require further characterization with techniques that are not as common as XPS, EXAFS or XANES. But we can consider the following: The XRD profile indicates that most of the analyzed material are nanoparticles, and that most of the precursor salts have been completely transformed into copper compounds. i.e., Since XRD yields no precursor residues, it can be assumed that most of copper(II) sulfate has been transformed into CuO, copper(II) nitrate and copper(II) complex. However, we cannot be sure that all the copper is part of a nanostructure.

2. Have you processed the nanoparticles after preparation to remove the CuSO4 salts? If not, then have you tested the antimicrobial activity of CuSO4 salts as control?

No, we did not process further the nanoparticle suspensions, they were used as obtained. Preliminary analyses show no inhibition of bacteria with CuSO4 (we have added representative image in the manuscript).

3. The Size distribution histogram is showing AgNPs instead of CuONPs (fig 1&2 C)

Thank you for the observation, the mistake was corrected.

4. UV-Vis peak of CuONPs is also not clear as both supernatant (red) and CuONPs (green) showing same peak (Fig1&2 A) (no shift in the peak).

The shift is almost imperceptible since the extract of Ganoderma and CuONPs peak at 290-300 nm.

5. E. coli growth is inhibited at 16.5ug/mL but again growth at 27.5 ug/mL. What could be reason for that?

We have revised our data and there was an error, it was corrected.

6. What is reason for the fluctuation in growth at different concentration of CuONPs in P. aeruginosa and S. aureus? 

It was observed that at low concentrations of CuONPs bacteria can grow, these fluctuations in %CFU are possibly the result of that (see Fig. 6A).

Reviewer 2 Report

In this study, Copper Oxide Nanoparticles (CuONPs) are created sustainably using a fungal extract. These minuscule, almost spherical particles exhibit promising antimicrobial activity against bacterial strains like E. coli, P. aeruginosa, and S. aureus while requiring minimal inhibitory concentrations. Remarkably, these nanoparticles permeate bacteria, as illustrated by microscopic analysis. On testing CuONPs' toxicity on mammalian cells, they largely exhibit safety, with a slight caveat for liver cells at higher concentrations. The compelling results imply that CuONPs could be a prospective contender for treating superficial infectious diseases - a hypothesis warranting further exploration. Overall the research is straight and clear, I do not have additional comment for this paper.

Author Response

Thank you for your positive comments.

Regards

Reviewer 3 Report

Manuscript entitled, “Antibacterial activity of biosynthesized copper oxide nanoparticles (CuONPs) using Ganoderma sessile” is the detail investigation providing imperative information regarding biosynthesized copper nanoparticles antibacterial efficacy. I believe with minor improvements this manuscript can be published in the reputed journal of mdpi-Antibiotics.

My comments are following below:

1.      Add few sentences in the introduction, ‘why copper oxide, zinc oxide and iron oxide are more into consideration than the gold and silver in biomedical application’.

2.      Add references in the section “Biocompatibility of CuONPs in mammalian cell lines”.

3.      Author mentioned that the Synthesized NPs using the supernatant (CuONPs-S) displayed a dark blue color, whereas the picture attached in Fig.1. is not depicting the same. Kindly, replace the picture showing nanoparticle suspension to validate your results.

4.      How author decided to choose the dosage concentration for toxicity assay.

5.      24.8 mV is the value of zeta potential written in discussion? Add Negative sign.

Please check the minor grammatical mistakes in the article thoroughly.

Author Response

Reviewer 3

1. Add few sentences in the introduction, ‘why copper oxide, zinc oxide and iron oxide are more into consideration than the gold and silver in biomedical application’.

Sentences added.

2. Add references in the section “Biocompatibility of CuONPs in mammalian cell lines”.

Reference added.

3. Author mentioned that the Synthesized NPs using the supernatant (CuONPs-S) displayed a dark blue color, whereas the picture attached in Fig.1. is not depicting the same. Kindly, replace the picture showing nanoparticle suspension to validate your results.

For clarity we have changed both pictures in UV-Vis graphs, also changed “dark blue color” for “blue-grey color”.

4. How author decided to choose the dosage concentration for toxicity assay.

We used similar concentrations used for bacterial inhibition. This was clarified in the text.

5. 24.8 mV is the value of zeta potential written in discussion? Add Negative sign.

Corrected, thank you.

Reviewer 4 Report

The manuscript titled "Antibacterial activity of biosynthesized copper oxide nanoparticles (CuONPs) using Ganoderma sessile" explores the antimicrobial and biocompatibility properties of CuO nanoparticles derived from Fungus supernatant and extract. This eco-friendly biosynthesis method offers potential economic benefits and novel insights in the field. The results highlight the use of Ganoderma sessile as a sustainable resource for producing small quasi-spherical nanoparticles ranging from 4.5 to 5.2 nm. The manuscript investigates the biological activity of the biosynthesized CuONPs against various pathogens, including E. coli, P. aeruginosa, and S. aureus, all demonstrating excellent minimum inhibitory concentrations. Additionally, the manuscript evaluates the ultrastructure and toxicity of the samples, showing their non-toxic effects at low concentrations. Overall, this manuscript holds promise for significant contributions to the scientific community.

However, there are some significant revisions that should be considered before accepting the manuscript for publication:

  1. The abstract provides a good overview of the study, but some improvements can be made to increase clarity and provide more specific details on the study's methodology and potential economic benefits.
  2. In the abstract, abbreviated concepts should also be accompanied by their corresponding abbreviations when they are first mentioned. For example, in the sentence "CuONPs showed antimicrobial activity against Escherichia coli, Pseudomonas aeruginosa, and Staphylococcus aureus," the abbreviations can be included in parentheses as follows: "CuONPs showed antimicrobial activity against Escherichia coli (E. coli), Pseudomonas aeruginosa (P. aeruginosa), and Staphylococcus aureus (S. aureus).
  3. Page 1-Page 2: The section on the implementation of biological methods and fungi as sustainable sources for NP production could be expanded to provide more details, particularly examples of the potential benefits and drawbacks of this approach. For example, you can refer to the following MDPI articles: https://doi.org/10.3390/ma16051798, https://doi.org/10.1039/D3NJ00131H.
  4. The introduction could benefit from referencing previous reports that highlight the main points, such as justifying the L6-8 (P2) by restating the benefits of green methods that don't require reducing agents. It can be mentioned that these methods act as both reducing and stabilizing agents due to the presence of antioxidant, antibacterial, and antibiotic compounds, including phytochemicals and polyphenolic compounds.
  5. In the objective section, the authors should emphasize the novelty of their study and highlight why they have chosen both fungal supernatant and extract, rather than other parts of the fungi. Additionally, authors should state the knowledge gap bridging the existing literature.
  6. Were Sections 2.1, 2.2, and 2.3 developed based on previous studies? Please provide the appropriate references to support these sections.
  7. In Section 2.3, the authors should justify the conditions of using a 3:1 precursor to reducing agent ratio and the determination of pH 10. It is recommended to refer to previous suggested papers that have extensively studied and reported on these specific conditions to support their justification.
  1. XRD and FTIR are both characterization tools commonly used for NPs. Therefore, the authors are suggested to combine sections 2.5 and 2.6 with section 2.4. Additionally, the authors can benefit from previous references to provide information on how data such as NP size and crystallinity were calculated using XRD technique.
  2. Authors should provide a brief description of the sample preparation for each of the characterization techniques used in their study.
  3. In Sections "2.7. Evaluation of Antibacterial Activity" and "2.6. ROS production in bacteria," authors should clearly mention the NP concentrations and the solvent in which they were prepared.
  4. Please double-check the section and subsection numbering. For example, check the numbering of Section “2.6. ROS production in bacteria” to ensure accuracy.
  5. In Page 3: Please double-check the sentence “cells were washed thrice with 200 μL of PBS 1× and incubated in darkness with 100 μl of DCFDA (20,70-dichlorofluorescein diacetate, 45 μM) (D6883 Sigma-Aldrich) for 60 min at 37°C.”
  6. Section 3: The average size of CuNPs-E is not included, and the statement regarding NP stability should be double-checked and supported with relevant literature. For example, Zeta potential between ±30 mV is considered highly stable.
  7. The XRD reveals a mixture phase of component-derived Cu, including oxides, nitrates, and hydroxides. How can the authors ensure the purity of these green-prepared NPs.
  8. The Zeta potential value of CuONPs-E was positive (24.8 mV) in the discussion but negative in the other parts of the manuscript. Please double-check and confirm which value is correct.
  9. In the discussion section, there are instances of verbosity and redundancy that have already been addressed in the introduction. Some parts may need to be moved to the introduction to streamline the content. However, I would recommend the discussion section focus on presenting the main results of the study and, if necessary, justifying those results based on the conditions used. Comparisons with existing literature should also be included.
  10. To enhance clarity, readability, and accuracy, information regarding parameter selection should be included in the materials and methodology section (as indicated in the previous comment).
  11. Regarding the antibacterial mechanism, additional structures and reactions can be found in the following references for further improvement: doi: 10.1007/s00253-023-12364-z and https://doi.org/10.1007/s12668-021-00851-4.   
  12. The conclusion could be expanded further to include specific recommendations for future studies and highlight potential challenges in the field. This would provide a comprehensive perspective on the research findings and suggest avenues for further exploration.
  13. The manuscript should undergo a thorough double-check to identify and correct any grammatical and spelling errors, such as unit separation from the mean, superscripts, and other formatting issues.

The manuscript should undergo a thorough double-check to identify and correct any grammatical and spelling errors.

Author Response

Thank you for your valuable comments/suggestions. Please see the attached document.

Regards

Round 2

Reviewer 4 Report

The authors are highly appreciated for having addressed all the points and questions raised. The manuscript is almost ready for publication. A quick reminder to double-check minor superscripts and subscripts, such as "cu+2" in line 407 (should be "Cu+2"), and unit separation in lines 173 and 184.